Evidence for the Cretaceous shark Cretoxyrhina mantelli feeding on the pterosaur Pteranodon from the Niobrara Formation

Hone David W.E. d.hone@qmul.ac.uk dwe_hone@yahoo.com 1
Witton Mark P. 2
Habib Michael B. 3
1 School of Biological and Chemical Sciences, Queen Mary University of London , London , United Kingdom
2 School of Earth and Environmental Sciences, University of Portsmouth , Portsmouth , United Kingdom
3 Keck School of Medicine, University of Southern California , Los Angeles , United States of America
Young Mark
Electronic publication date: 2018 Dec 14
Publication date: 2018
Volume: 6
Electronic Location ID: e6031
Received 2018 Sep 18; Accepted 2018 Oct 30
Copyright: ©2018 Hone et al.
Copyright year: 2018
Copyright holder: Hone et al.
License: This is an open access article distributed under the terms of the Creative Commons Attribution License, which permits unrestricted use, distribution, reproduction and adaptation in any medium and for any purpose provided that it is properly attributed. For attribution, the original author(s), title, publication source (PeerJ) and either DOI or URL of the article must be cited.
License URL: https://creativecommons.org/licenses/by/4.0/

Keywords: Pterosaur, Palaeoecology, Behaviour, Carnivore-consumed, Predation, Scavenging

Funding: The authors received no funding for this work.

==============================
A cervical vertebra of the large, pelagic pterodactyloid pterosaur Pteranodon sp. from the Late Cretaceous Niobrara Formation of Kansas, USA is significant for its association with a tooth from the large lamniform shark, Cretoxyrhina mantelli. Though the tooth does not pierce the vertebral periosteum, the intimate association of the fossils—in which the tooth is wedged below the left prezygapophysis—suggests their preservation together was not mere chance, and the specimen is evidence of Cretoxyrhina biting Pteranodon. It is not possible to infer whether the bite reflects predatory or scavenging behaviour from the preserved material. There are several records of Pteranodon having been consumed by other fish, including other sharks (specifically, the anacoracid Squalicorax kaupi), and multiple records of Cretoxyrhina biting other vertebrates of the Western Interior Seaway, but until now interactions between Cretoxyrhina and Pteranodon have remained elusive. The specimen increases the known interactions between large, pelagic, vertebrate carnivores of the Western Interior Seaway of North America during the Late Cretaceous, in addition to bolstering the relatively small fossil record representing pterosaurian interactions with other species.

Introduction

Pteranodon is a large pterodactyloid pterosaur from the Late Cretaceous (Coniacian-Campanian) of North America with an estimated maximum wingspan of 7.25 m (Bennett, 2001). The genus was among the first pterosaurs reported from North America (Marsh, 1876)—see (Bennett, 2001; Witton, 2010 for context of its discovery) and has become one of the best known flying reptiles thanks to a representation of over 1,100 specimens—the highest sample size for any pterosaur genus. Although most specimens are incomplete and crushed, every component of its osteology is known and has been described in detail (Eaton, 1910; Bennett, 1991; Bennett, 1994; Bennett, 2001; Bennett, 2018; Bennett & Penkalski, 2018). As a result of the number of available specimens, its long research history and comprehensive documentation, the genus has become a cornerstone of pterosaur research. Pteranodon has been an important animal for understanding pterosaur flight (Hankin & Watson, 1914; Bramwell & Whitfield, 1974; Stein, 1975), the evolution of giantism in flying animals (Witton & Habib, 2010), pterosaur ontogeny (Bennett, 1993), and palaeoecology (Bennett, 2001; Witton, 2018).

The majority of Pteranodon specimens are known from the Late Cretaceous Niobrara Formation from Kansas, USA, a marine deposit created by the Western Interior Seaway, though other specimens also occur in additional formations in Wyoming and South Dakota (Bennett, 1994; Bennett, 2001). Niobrara specimens of Pteranodon occur in localities that were hundreds of kilometres from the palaeocoastline and this, along with a number of aspects of functional anatomy, has seen the genus long interpreted as a seagoing, pelagic animal (e.g.,  Bennett, 2001; Witton, 2013, p. 179).

Pteranodon was likely an important component of the Western Interior Seaway ecosystem. It seems to have been relatively abundant, being known from a large number of fossils and making up some 97% of Niobrara Formation pterosaur finds. It was also a large animal—Bennett (1992) identified a bimodal size distribution among the Pteranodon sample where two thirds of individuals were c. 3.5 m in wingspan, and the remaining third were much larger, some exceeding 6 m across the wings (Bennett, 2001). Larger specimens likely exceed the masses of any flying bird, extant or extinct, with estimated body masses of 35–50 kg for animals of 6 m wingspan (Paul, 2002; Witton, 2008; Henderson, 2010), compared to 21.9–40.1 kg in the largest fossil flying birds, the pelagornithids ( Mayr & Rubilar-Rogers, 2010; Ksepka, 2014). Pteranodon populations may therefore have been major consumers in the Western Interior Seaway ecosystem, as well as potentially sources of food for other animals.

However, our understanding of interactions between Pteranodon and other Seaway taxa are limited. As with other pterosaur species, few Pteranodon fossils preserve remains of ingested content and they only rarely record evidence of consumption by other animals (Witton, 2018). Regurgitated fish are preserved in the gular region of one Pteranodon specimen (Brown, 1943; Bennett, 2001; Bennett, 2018) and some palaeoecological significance has been ascribed to small fish vertebrae found in association with Pteranodon fossils (Bennett, 2001; Hargrave, 2007; Ehret & Harrell Jr, 2018). Biting traces on Pteranodon elements show that some individuals were eaten by the anacoracid shark Squalicorax kaupi as well as a saurodontid fish, most likely Saurodon or Saurocephalus (Witton, 2018; Ehret & Harrell Jr, 2018). The record of pterosaur ecological interactions is sufficiently sparse that any fossilised interactions with other species should be put on record, so we hereby report on a series of Pteranodon cervical vertebrae, LACM 50926, associated with a tooth of the lamniform shark Cretoxyrhina mantelli. This is first documented occurrence of this large shark interacting with any pterosaur.

Systematic Nomenclature

The taxonomy of Pteranodon is a matter of recent dispute. For the last two decades most workers have followed the treatment of the genus outlined by Bennett (1994), who made a case for reducing the 11 binomials associated with Pteranodon (excluding those names related to Nyctosaurus) to two sexually dimorphic chronospecies: the older Pteranodon sternbergi and the younger P. longiceps. In this scheme, the skulls of these species are distinguished by details of their cranial crests, and (more tentatively) occiput orientation and mandibular ramus depth. Postcranial bones of these specimens are nearly identical and of little taxonomic utility (Bennett, 1994). More recently, Kellner (2010) argued that Pteranodon sensu Bennett (1994) actually comprised four species in three genera. While agreeing with Bennett (1994) that all ‘historic’ Pteranodon species were problematic excepting longiceps and sternbergi, Kellner (2010) created a multi-taxic pteranodontid assemblage for the Niobrara specimens comprising Pteranodon longiceps, Geosternbergia (rather than Pteranodon) sternbergi, and two novel species, Geosternbergia maiseyi and Dawndraco kanzai. These taxa are primarily distinguished by headcrest morphology and details of the posterior skull, as well as finer stratigraphic divisions of the Niobrara Formation (Kellner, 2010) than the broader ‘upper’ and ‘lower’ divisions of the Smoky Hill Chalk Pteranodon fauna recognised by other workers (e.g.,  Bennett, 1994; Everhart, 2005; Carpenter, 2008). Subsequent criticism has questioned the validity of the proposed differences between at least Dawndraco and Pteranodon sensu Bennett, noted incongruence between the stratigraphic divisions signified by Kellner (2010) against other Niobrara Formation taxa, as well as the lack of statistical support for splitting Pteranodon into multiple genera, compared to the strong statistical support for Bennett’s interpretation (Martin-Silverstone et al., 2017; Acorn et al., 2017). We thus follow several other works (Witton, 2013; Witton, 2018; Bennett & Penkalski, 2017; Bennett, 2018) in retaining Bennett’s (1994) treatment of Pteranodon here. Note that discussion of Pteranodon taxonomy is ongoing, however (Brandão & Rodrigures, 2018).

Materials and Methods

LACM 50926 (Los Angeles County Museum of Natural History, USA) is a specimen of Pteranodon mounted in a large glass case for public display at the Los Angeles County Museum of Natural History, which makes it difficult to access directly (Fig. 1). The specimen has a large Cretoxyrhina mantelli tooth intimately associated with the fourth cervical vertebra (Fig. 2). Parts of the mount are genuine, well preserved fossils showing only limited crushing compared to many specimens of the genus. However, several elements are reconstructed to replace missing parts and the mount is a composite of material from several individuals (see Bennett, 1991; Bennett, 2001), as is obvious from bone size discrepancies between several neighbouring elements (Fig. 1). SC Bennett (pers. comm., 2016) also notes material accessioned under this number (much of it in collections space and not in the exhibit mount) includes three mandibular rami, confirming the multi-individual nature of this specimen. An alternate specimen number (65218) occurs on the mandible and the cervical bearing the shark tooth, but this cannot be seen on other elements. This may indicate that the mandible and cervical were associated when discovered. Bennett (2001) was able to identify many of the LACM 50926 forelimb elements as belonging to a single individual, although there are no records to indicate which parts of the mounted specimen might relate directly to the cervical series. The preservation quality and size of the vertebrae correspond well to the other elements (including the forelimb bones) and this implies that LACM 50926 may represent much of a skeleton. However, the absence of both anteriormost and posterior cervical vertebrae means no anatomical continuity links the 50926 vertebrae with the rest of the material, and their association to the rest of the skeleton cannot be confidently assumed.

Figure 1 Mounted Pteranodon and close up of the neck.

(A) mounted Pteranodon sp. skeleton LACM 50926 on display in the Los Angeles county museum with highlighted section of the vertebrae shown below; (B) Close up of the vertebral series and shark tooth (indicated by an arrow). Cervical vertebrae III–VII are indicated. Scale bar is 50 mm—this is an approximate value based on published measurements of the vertebrae. Image credit: (A) Stephanie Abramowicz, courtesy Dinosaur Institute, Natural History Museum of Los Angeles County, (B) David Hone.

Figure 2 Two close up views of the Cretoxyrhina mantelli tooth with tracings.

(A) left dorsolateral view; (B) left dorsoventral view showing its intimate association with cervical vertebra IV. The tooth is highlighted in medium grey, the 4th cervical vertebra in pale grey and the 5th cervical in dark grey. Abbreviations: ns neural spine, prz prezygopophysis, psz postzygopophysis, st shark tooth. Image credit: David Hone.

Notes held at the LACM show that the specimen was collected in 1965 by M.C. Bonner from Niobrara Chalk 23, Niobrara Formation, Logan County, Kansas. Bennett (1991) refers to two specimens under this number (LACM 50926 and 50926 “A”) and concurs with this locality, adding that they were collected between Marker Units 14 and 19. This makes a Santonian age likely for LACM 50926 (Hattin, 1982; Bennett, 1994).

Description

The anatomy of Pteranodon has been described in detail elsewhere (Bennett, 2001) and we will therefore focus exclusively on the association between the shark tooth and pterosaur material. The cervical vertebra bearing the shark tooth is preserved in contact with two other cervicals as a series of three elements. Thus, within the composite context of the LACM specimen, these vertebrae at least can be safely considered part of a single individual. The cervicals are preserved with contact between the successive post- and prezygapophyses. These are identified by Bennett (2001) as cervical vertebrae 4–6, and he also identified a preceding, though not articulated, cervical in the LACM 50926 mount as a cervical 3. The vertebrae retain some three-dimensionality, although they are partially crushed at an oblique angle, shearing them along their midline such that the left sides are depressed and right sides elevated (Figs. 1 and 2). The neural spine is missing (now restored) from cervical 4 and parts of the neural spines of cervicals 5 and 6 are damaged. Damage to the bone cortex reveals the internal structure of the bones in all three vertebrae.

The centrum lengths of the three cervical vertebrae in the series have been measured as 69.0, 77.8, and 71.5 mm respectively (Bennett, 2001). Based on comparisons to other specimens (see Bennett, 2001) the individual represented by LACM 50926 had a c. 5 m wingspan, and was presumably therefore osteological adult or near adult in size. The embedded shark tooth is approximately 24 mm in total height (root plus crown) (this was measured from photographs as it was impossible to measure the tooth given its location and the mount of the specimen), subtriangular in shape and highly compressed labiolingually. A wide, lunate root is formed from two obtusely angled, swollen root lobes. The termination of the left lobe (viewed from lingual aspect) forms a broad, somewhat rounded surface, but the termination of the right lobe is missing (Fig. 2). The crown is swollen on the labial surface, c. 12 mm long (measured from the base of the root to apex of the crown), almost symmetrical but not significantly recurved with respect to the root. No serrations are apparent on the tooth crown but the lateral and medial crown edges are somewhat worn, with chipped margins. The tooth enameloid is bright white with grey to brown patches, and the root is pale grey-brown and close in colour to that of the pterosaur elements.

The tooth lies between the left prezygapophysis of cervical 4 and the centrum. Although it appears that the tooth is wedged or has cut into the base of the prezygapophysis and centrum it actually lies medial to the prezygapophysis and does not directly contact this projection. The tooth is preserved at a shallow angle to the long axis of the vertebra (though this may reflect the crushing of the specimen rather than its original orientation) and the apex of the crown faces posteriorly and ventrally with respect to the vertebral corpus. The tooth does not penetrate the centrum, but the tip of the tooth is in contact with its surface.

Results

Taxonomic identities

The composite nature of LACM 50926 complicates discussions of its affinities, but there is no doubt that the specimen can be referred to Pteranodon given its provenance and matching anatomy to this pterosaur (Eaton, 1910; Bennett, 2001). Identification to species level is more problematic as Pteranodon taxonomy is exclusively informed by the posterior skull region (e.g., Eaton, 1910; Bennett, 1994; Kellner, 2010), and the vertebra is not associated with any skull material. Following Bennett’s (1994) tentative suggestion that P. sternbergi may have a shallower mandible than P. longiceps we compared the LACM 50926 mandibular ramus with specimens referred to these species. However, we were unable to determine a significant match with either taxon. Hargrave (2007) suggested that the tomial margins of posterior P. longiceps mandibles are curved, and this morphology is present in the LACM 50926 mandible. However, while we agree this can be seen in some P. longiceps (e.g., YPM 2594 - YPM, Yale Peabody Museum, USA) it does not seem to be a universal trait (e.g., YPM 1177).

The recovery of LACM 50926 from marker units 14–19 of Hattin’s (1982) Smoky Hill Chalk stratigraphy gives it provenance among younger Niobrara beds yielding Pteranodon longiceps rather than P. sternbergi (Bennett, 1994; Carpenter, 2008), although (Kellner, 2010) argues that species more closely related to P. sternbergi than P. longiceps may persist into younger deposits). This indicates that LACM 50926 probably represents P. longiceps but, in lieu of diagnostic fossil material, we regard the specimen as Pteranodon sp.

A number of medium- to large-sized, sharp-toothed sharks are known from the Niobrara Formation, and they have left an extensive record of tooth traces and shed teeth among other vertebrates of the Smoky Hill Chalk Member (Everhart, 2005). The Niobrara species best known for leaving bite traces is Squalicorax kaupi, but this identification can be excluded for the LACM tooth because it lacks the asymmetrical crown and serrations characterising the dentition of this genus (e.g., see Everhart, 2005; Becker & Chamberlain, 2012). The tooth is a good march for the large lamniform shark Cretoxyrhina mantelli (Fig. 3), which has subtriangular, relatively broad, symmetrical and short crowns without serrated margins (e.g., Shimada, 1997; Siverson & Lindgren, 2005, their fig. 2; Bourdon & Everhart, 2011). In particular, the morphology of the tooth in LACM 50926 matches teeth recovered from anterior positions of Cretoxyrhina jaws (Shimada, 1997; Bourdon & Everhart, 2011, their figs. 2, 5). The identification of the shark tooth as belonging to Cretoxyrhina was also independently made by Konuki (2008) and Witton (2018). Comparison of the LACM tooth size with a superb C. mantelli skeleton, FHSM VP-2187 (Shimada, 1997), suggests that the individual was c. 2.5 m long. This is little more than one third of length of the largest known individuals of this species (Everhart, 2005).

Figure 3 Cretoxyrhina mantelli anterior teeth.

Tracing of Cretoxyrhina mantelli anterior teeth from Bourdon & Everhart (2011), their figure 5, mirrored from their original). (A) position 3 in the jaw; (B) position 4; (C) LACM 50926 tooth. The bases of the teeth are shaded in pale grey and the enamel is dark grey. Image credit: David Hone.

Discussion

Significance of association of Pteranodon and Cretoxyrhina

Ecological interactions between pterosaurs and other species are rarely represented in fossil specimens, despite vast increases in pterosaur specimen numbers in recent years (Witton, 2018). Data on diet from stomach contents is sparse, limited to a handful of taxa known to have eaten fish (e.g., Eudimorphodon—Wild, 1978, Pteranodon, Rhamphorhynchus—Wellnhofer, 1991). Coprolites are also scarce, with only one record for pterosaurs known to date (Hone et al., 2015). A number of animals are recorded as pterosaur consumers, including fish (e.g., Frey & Tischlinger, 2012), dinosaurs (e.g., Hone et al., 2012), Crocodyliformes (Vremir et al., 2013) and possibly plesiosaurs (Cicimurri & Everhart, 2001), although also see (Witton, 2018), but they remain very rare despite the good fossil records of these ‘consumer’ taxa. Thus, this additional potential record of a pterosaur-carnivore association is significant.

The taphonomic history and association of LACM 50926 is unknown so it is difficult to draw firm conclusions about the action that left the shark tooth in situ. However, we rule out abiotic association of the pterosaur and shark tooth for several reasons: (1) embedded Cretoxyrhina teeth and feeding traces are known from numerous Smoky Hill vertebrate fossils, and are widely interpreted as related to feeding behaviour (Shimada, 1997; Everhart, 2004; Everhart, 2005); (2) although isolated Cretoxyrhina teeth are common fossils in the Smoky Hill Chalk Member (Everhart, 2005), its teeth have not been reported in association with any Pteranodon fossils in the past, despite the large sample size of this pterosaur and the fact that other fish remains (e.g., vertebrae) are not uncommonly associated with their remains (Bennett, 2001; Hargrave, 2007); (3) the spatial relationship between the tooth and the vertebra is complex and intimate, and unlike that expected to have occurred by chance association in a low energy deposit such as the Niobrara Chalk. We thus prefer an interpretation of the tooth becoming associated with the vertebra though the biting action of a small Cretoxyrhina.

We were unable to find additional indications of bite traces on LACM 50926. There is a small and almost perfectly circular puncture on the neural arch of cervical four, behind the left prezygapophysis but this is most likely a preparation trace or damage derived from a previous museum mount. The damaged and missing neural spines of the cervical series may be linked to the shark bite, but other pterosaur fossils show that these elements are prone to damage and/or poor preservation, so other causes cannot be excluded.

Cretoxyrhina was a large (up to 7 m in length) and powerful carnivore, perhaps one of the top predators of the Smoky Hill Chalk fauna (Everhart, 2005). Shimada (1997) compared its likely ecological feeding guild to larger modern species of lamnid and carcharhinid sharks, and there is fossil evidence that it consumed a variety of large vertebrates including mosasaurs, plesiosaurs and large teleost fish (Shimada, 1997; Everhart, 2004; Everhart, 2005). LACM 50926 is the first palaeoecological link between this shark genus and a pterosaur however, this rarity perhaps reflecting the relatively delicate nature of pterosaur skeletons against the evident bite force of Cretoxyrhina. Extremely hollow bones such as those characterising most of the Pteranodon skeleton are especially prone to failure against buckling forces (Currey, 2002) and likely broke easily under strong bites from large predators.

Both Bennett (2001) and Hargrave (2007) have noted that Pteranodon may have been consumed destructively by large aquatic carnivores. Predator targeting of their relatively muscular torsos might explain why wing skeletons (which had considerably less soft-tissue, see Bennett, 2003) are the most common form of associated pterosaur fossil in the Smoky Hill Chalk Member. Articulated wings are also common in the Late Jurassic Solnhofen fauna where this may reflect decay and the loss of wings from floating pterosaur corpses (Beardmore, Lawlor & Hone, 2017), although this is not mutually exclusive with the effects of predation and scavenging. Witton (2018) noted that, to date, only the larger, more robust elements—limb bones and neck vertebrae—of larger pterosaur species are known to preserve embedded teeth and speculated that small pterosaurs and more gracile pterosaur bones were probably too easily destroyed to record evidence of carnivore bites. It may be that pterosaurs were not rare dietary components of Cretoxyrhina or other animals, but that their anatomy precludes common fossilisation of evidence for these acts.

Figure 4 Restored scene of Cretoxyrhina attacking Pteranodon.

Life reconstruction of a c. 2.5 m long breaching Cretoxyrhina mantelli biting the neck of a 5 m wingspan Pteranodon longiceps, a scene inspired by LACM 50926. The predatory behaviour of this scene is speculative with respect to the data offered by the specimen, but reflects the fact that Cretoxyrhina is generally considered a predatory species, the vast weight advantage of the shark against the pterosaur (see text), and the juvenile impulse of the artist to draw an explosive predatory scene. Image credit: Mark Witton.

There is limited potential for knowing whether the LACM 50926 association reflects predatory or scavenging behaviour from Cretoxyrhina. Pteranodon is widely considered to have been a pelagic pterosaur species which foraged for small aquatic prey by means of dip-feeding, fishing from an alighted position on the water surface or diving after food (Wellnhofer, 1991; Bennett, 2001; Witton, 2013; Witton, 2018). Adaptations to aquatic launch (identified by Habib & Cunningham, 2010) are apparent in Pteranodon and suggest that it may have routinely entered (and thus needed to launch from) bodies of water. Thus, there are good reasons to think living Pteranodon could have been within reach of predatory sharks, and the likely pterodactyloid floating posture places their head and neck close to the waters’ surface (Hone & Henderson, 2014). Various modern seabirds are predated by pelagic predators, including sharks (Wetherbee, Cortés & Bizzarro, 2004; Johnson et al., 2006), and we cannot exclude this possibility for the LACM Pteranodon. Witton (2018) noted that even moderately-sized sharks, akin to the 2.5 m long Cretoxyrhina indicated by the LACM tooth, would vastly outweigh the largest Pteranodon (35–50 kg—see Paul, 2002; Witton, 2008; Henderson, 2010 for Pteranodon mass estimates), and we have little doubt that such predators could subdue these pterosaurs if they caught them (Fig. 4). Conversely, Pteranodon likely had a relatively low body density and their carcasses may have floated for sustained periods (Hone & Henderson, 2014). This would make them obvious targets for scavenging marine animals. Ultimately, LACM 50926 preserves no evidence to falsify any of these hypotheses.

Evidence of the anacoracid shark Squalicorax consuming Pteranodon is known in the Niobrara (e.g., KU 972 - KU, Kansas University, USA; YPM 2597, YPM 42810—SC Bennett, pers. comm., 2016), and recent finds of Mooreville Chalk Formation Pteranodon also have bite marks attributed to Squalicorax kaupi (RMM 3274 and ALMNH 8630) Ehret & Harrell Jr (2018). This body of evidence, augmented with the Cretoxyrhina-Pteranodon association described here, and the recovery of fish remains within the gular region of Pteranodon specimens (Brown, 1943; Bennett, 2001; Bennett, 2018) makes the trophic interactions of Pteranodon well understood compared to most other pterosaurs (Witton, 2018). However, such finds are still relatively rare occurrences—these seven associations are less than 1% of the >1,100 specimens of Pteranodon on record. In contrast, at least ten palaeoecologically significant fossil associations are known for the Late Jurassic Solnhofen pterosaur Rhamphorhynchus muensteri (including five associations with the carnivorous fish Aspidorhynchus acutirostris (e.g., Frey & Tischlinger, 2012) and four examples of consumed items (Witton, 2018). There are perhaps 150 specimens of Rhamphorhynchus in public collections, suggesting that recording of palaeoecological events is several times higher than in Pteranodon (>6%) despite a considerably smaller sample size. The taphonomic factors contributing to this difference may be worthy of further study.

We thank Luis Chiappe and Maureen Walsh for access to the specimen and information on its history and Stephanie Abramowicz for access to archival photographs of the specimen. We would like to thank Steffi Klug for assistance on identification of the shark tooth, Dana Ehret and Chris Bennett for discussions on the composite mount, bite marks on Pteranodon bones and providing key literature. Dana Ehret and Chris Bennett are further thanked for their reviews of this paper, and Mark Young is thanked for his handling of the manuscript; all three helped improve this work.

Additional Information and Declarations

Competing Interests

Author Contributions

Data Availability

The authors declare there are no competing interests.

David W.E. Hone conceived and designed the experiments, performed the experiments, analyzed the data, contributed reagents/materials/analysis tools, prepared figures and/or tables, authored or reviewed drafts of the paper, approved the final draft.

Mark P. Witton conceived and designed the experiments, performed the experiments, analyzed the data, prepared figures and/or tables, authored or reviewed drafts of the paper, approved the final draft.

Michael B. Habib conceived and designed the experiments, performed the experiments, authored or reviewed drafts of the paper, approved the final draft.

The following information was supplied regarding data availability:

The research in this article did not generate any data or code. The specimen is housed at the Los Angeles County Museum of Natural History, USA: specimen ID: LACM 50926.

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
