# Peer review of "Evidence for the Cretaceous shark Cretoxyrhina mantelli feeding on the pterosaur Pteranodon from the Niobrara Formation"

_PeerJ, doi:10.7717/peerj.6031_

## Round 0.1 · original submission · Minor Revisions

Dear authors,

I have accepted the reviewers' recommendation of 'minor revisions'.

I look forward to receiving your revised manuscript.

·

Basic reporting

This manuscript is well thought out, and written well aside from a few small typos. It was easy to read, clear and concise. Within my attached review I made comments with regards to English and grammar when needed. One suggestion I have for the authors is to state where and when the specimen is from in the abstract. It is not very clear and would help in future literature searches.

The literature references were appropriate and well researched. I recommend the authors reviewing Ehret and Harrell, 2018 (which was just published) and have added a few other suggested papers, but they are not necessary per se. The figures are done well and are consistent with the content of the manuscript.

Artist's reconstruction of the potential event, while somewhat fantastical, is wonderful. The author's fully admit that artistic license was taken.

Experimental design

This original manuscript reviews the predation of a pterosaur which is preserved in a specimen from Kansas. The specimen is mentioned on the Oceans of Kansas website, and has been cited in other research papers but it has not been formally researched and described. Due to the fragile nature of pterosaur bones, preservation is not common, in particular specimens with evidence of predation. The authors do a good job describing the specimen and discussing how it came to be. The specimen is likely a composite, however, abundant evidence and citations are given to demonstrate that the vertebrae and shark tooth are a truly associated specimen. In my attached review, I suggest that the authors further investigate the orientation of the shark tooth in the specimen. Thinking about the orientation of the tooth while it was in the shark's mouth, and the mechanics of how the event likely occurred, it lends more support to the authors' conclusions. I also suggest the authors familiarize themselves more with the common terminology of shark teeth. Portions of the description were somewhat confusing.
With the few numbers of pterosaur bones present with predation/scavenging marks, this research is an important contribution to the field.

Validity of the findings

The authors describe the specimen in full and are conservative with their hypothesis. Their conclusions are well founded and researched.

Additional comments

See other comments in marked up manuscript. Aside from a few suggestions regarding references and relating tooth orientation to feeding/scavenging event, I am pleased with this report.

·

Basic reporting

Clear and unambiguous, professional English used throughout.

Literature references, sufficient field background/context provided.

Self-contained with relevant results to hypotheses.


Meets standards.

Experimental design

They describe the specimen, compare it to similar specimens, etc. Interpret the results.


Meets standards.

Validity of the findings

All seems valid, Cretoxyrhina bit on Pteranodon, and why wouldn't it. The only questionable thing is the acrobatic shark snatching the pterosaur out of the air, which the authors do not suggest was normal feeding mode of Cretoxyrhina.

Additional comments

See that attached pdf for more detailed comments.

---

## Round 0.2 · accepted · Accept

Dear authors,

Many thanks for your revised manuscript.

I hope you continue to use PeerJ as your publication venue in the future.

#